# Determinants of COVID-19 Vaccinations among a State-Wide Year-Long Surveillance Initiative in a Conservative Southern State

**DOI:** 10.3390/vaccines10030412

**Published:** 2022-03-09

**Authors:** Lídia Gual-Gonzalez, Maggie S. J. McCarter, Kyndall Dye-Braumuller, Stella Self, Connor H. Ross, Chloe Rodriguez-Ramos, Virginie G. Daguise, Melissa S. Nolan

**Affiliations:** 1Department of Epidemiology and Biostatistics, University of South Carolina, Columbia, SC 29208, USA; lidiag@email.sc.edu (L.G.-G.); msm6@email.sc.edu (M.S.J.M.); kyndallb@email.sc.edu (K.D.-B.); scwatson@mailbox.sc.edu (S.S.); chross@email.sc.edu (C.H.R.); cr21@email.sc.edu (C.R.-R.); daguisvg@dhec.sc.gov (V.G.D.); 2South Carolina Department of Health and Environmental Control, Columbia, SC 29201, USA

**Keywords:** vaccine hesitancy, principal component analysis, GIS, survival analysis, COVID-19, SARS-CoV-2

## Abstract

By the end of 2021, the COVID-19 pandemic resulted in over 54 million cases and more than 800,000 deaths in the United States, and over 350 million cases and more than 5 million deaths worldwide. The uniqueness and gravity of this pandemic have been reflected in the public health guidelines poorly received by a growing subset of the United States population. These poorly received guidelines, including vaccine receipt, are a highly complex psychosocial issue, and have impacted the successful prevention of disease spread. Given the intricate nature of this important barrier, any single statistical analysis methodologically fails to address all convolutions. Therefore, this study utilized different analytical approaches to understand vaccine motivations and population-level trends. With 12,975 surveys from a state-wide year-long surveillance initiative, we performed three robust statistical analyses to evaluate COVID-19 vaccine hesitancy: principal component analysis, survival analysis and spatial time series analysis. The analytic goal was to utilize complementary mathematical approaches to identify overlapping themes of vaccine hesitancy and vaccine trust in a highly conservative US state. The results indicate that vaccine receipt is influenced by the source of information and the population’s trust in the science and approval process behind the vaccines. This multifaceted statistical approach allowed for methodologically rigorous results that public health professionals and policy makers can directly use to improve vaccine interventions.

## 1. Introduction

As of December 2021, the COVID-19 pandemic has resulted in over 350 million cases and 5.36 million deaths worldwide, and approximately 54,9 million cases and 827,823 deaths in the U.S. alone [1]. This situation has resulted in significant morbidity and has greatly impacted public health preparedness and response due to its novel nature. The first response was to implement a generalized lockdown at the beginning of the pandemic and restrict mobility to curb the spread of SARS-CoV-2 [2,3,4]. Throughout the pandemic, although resources were scarce, new tools and strategies were developed. One of these strategies was the development of guidelines to prevent the spread of COVID-19, which included promoting social distancing, handwashing, mask wearing, frequent testing and vaccination [5]. These guidelines are not always well received by the public, which posed a challenge for successfully preventing disease spread [5]. Meltzer et al. found that adherence to public health measures was related to worry about contracting COVID-19 [5], and in a study of Reader et al., at least 17–21% of the respondents were not likely at all to wear a mask [6]. This previous research indicates that for successful public health responses, we must take population behaviors and disease knowledge into consideration [5,6]. Strict public health measures have not always been effective, and level of adherence is dependent on the population’s trust and knowledge, and their information sources [3,5].

Vaccines against SARS-CoV-2 are an invaluable tool to help prevent severe disease and to prevent swamping the health care system. Vaccine reluctancy or refusal, also known as vaccine hesitancy, has been deemed a global concern [7]. Vaccine hesitancy has exacerbated multiple outbreaks and has led to dire economic consequences in the past [8,9]. The current pandemic has seen surges in hesitancy and vaccination refusal, and vaccine hesitancy is one of the greatest barriers in pandemic control [7]. This hesitancy has resulted in relatively low vaccination rates, contributing to increased burden of the new SARS-CoV-2 variants (i.e., B.1.1.7, delta, omicron, etc.). The evolution of variants has diminished the protection of currently available vaccinations, maintained the advent of primary infections, and led to a surge of reinfections among unvaccinated persons, and has brought breakthrough infections among vaccinated persons [10].

In South Carolina, the first case of COVID-19 was detected on 4 March 2020. Between March 2020 and December 2021, South Carolina has accumulated approximately 940,000 confirmed cases and over 14,000 confirmed deaths despite the implementation of mitigation measures and the promotion of vaccines [11]. Vaccinations in South Carolina began on 14 December 2020, with doses being given to those eligible in phase 1-A (i.e., health care workers, first responders, etc.) followed by an age-group-escalated vaccination roll-out. Following the emergence of the delta variant, recommendation for a COVID-19 booster shot was extended to all individuals 18 and over in November 2021. In early 2022, omicron replaced delta as the dominant variant in South Carolina [12,13].

As of December 2021, 2.8 million doses have been administered in South Carolina (Figure 1), with 59.3% of the eligible population having received at least one dose and 51.2% with a completed vaccine series [11]. These percentages are relatively low when compared to national averages, highlighting the need to understand and combat vaccine hesitancy in this population.

In October 2020, the Sampling and Testing Representative Outreach for Novel coronavirus Guidance (SC STRONG) initiative was established as a collaborative project between the South Carolina Department of Health and Environmental Control (SC DHEC) and the University of South Carolina to respond to the spread of COVID-19 in the state. This project was designed as a state-wide COVID-19 surveillance strategy, for a one-year period (Fall 2020 to Fall 2021), offering testing and a complementary health survey, as previously described [16]. Almost 15,000 residents participated in this sampling and health survey initiative, affording the opportunity to analyze vaccine hesitancy and refusal temporarily and geospatially during a formidable part of the SARS-CoV-2 pandemic. South Carolina is within the bottom ten states for poor vaccination rates in the United States [17] and serves as a representative population for conservative COVID-19 anti-vaccination residents nationally. Given the complexity of COVID-19 anti-vaccination sentiment, we applied three distinct statistical methodologies to assess COVID-19 vaccination determinants to guide rigorous public health response.

## 2. Materials and Methods

### 2.1. Participant Sample

Using population-proportionate-to-size cluster sampling method for the first two testing rounds (October 2020 through December 2020 and January 2021 through February 2021) and simple random sampling for the last two rounds (May 2021 through June 2021 and August 2021 through September 2021), we selected 750,063 South Carolina residents 18 years or older over the four different testing rounds. Sampling methods were changed half-way through the surveillance initiative to allow for greater recruitment. These residents were selected from a third-party direct-mail marketing listserv (MailersHeaven, Valencia, CA, USA). The selected participants were sent invitation letters to participate in the SC STRONG initiative along with a household member aged 5 years or older. The letters invited participants to complete an online health survey or over the phone and to provide biological samples for SARS-CoV-2 PCR and antibody testing (not used for this analysis). Data cleaning methods included internal data point validation checks, coding, outlier removal, and typographical error correction. Cleaned survey responses were analyzed using three different statistical methods to evaluate vaccine hesitancy and vaccination status. Sample size varied by analysis depending on the variables involved.

### 2.2. Principal Component Analysis

Participants who completed the survey prior to 7 January were not asked key questions about vaccine/government trust that we wished to include in the PCA. As a result, the PCA analysis was performed only on the 5692 complete survey responses collected after 7 January. PCA was performed to reduce the dimensionality within our data and to understand latent factors determining vaccine hesitancy. Because our data contain a mixture of ordinal and Likert variables, along with other data types, a PCA using multiple correlation types was performed using R studio’s COR option “mixed”. The following variables were included in our final principal component analysis model: feelings regarding trust in the science of the COVID-19 vaccine and trust in the government, various actions to control spread of the virus, working environments, ethnicity, income, gender, age, sources from which individuals receive their information regarding the COVID-19 pandemic, feelings of stress or sadness during the pandemic, and if close friends or family have ever contracted COVID-19.

Each principal component is a linear combination of the original variables. The loadings associated with each principal component are the coefficients involved in this linear combination. A positive (negative) loading indicates a positive (negative) association between the principal component and the variable in question, with larger magnitude loadings corresponding to stronger relationships. Variables that have absolute loadings values of 0.40 or greater are considered to have a moderate contribution and will be highlighted in our analysis. Analyses were performed using R Software version 1.4.1103 and the “Psych” package [18].

### 2.3. Survival Analysis

Outcome Assessment. The primary outcome of interest was vaccination status (vaccinated/unvaccinated) and time to vaccine receipt in days, using the first dose of vaccine receipt for each participant. Time = 0 was set to 14 December 2020, based on the first date that the COVID-19 vaccine was available in South Carolina (Figure 1). The date of the survey (observation time) was utilized for individuals who did not receive the vaccine (censored). Any recorded dates of vaccine receipt that were impossible (before 14 December 2020) or those who indicated they received the vaccine, but did not provide a date, were eliminated. Because the SC Strong survey was administered during the staggered vaccine rollout plan, some individuals who intended to receive the vaccine but were not yet eligible were included. In order to account for this difference, individuals were asked if they intended to receive the COVID-19 vaccine and when: as soon as they are eligible, in a few weeks, 1–3 months, Fall 2021, or if they planned to wait longer. This variable was used to calculate the date of receipt of vaccine for those who would receive the vaccine but had not had the opportunity yet. Calculations were set as: eligibility date based on the SC DHEC rollout phases and comorbidity status, an additional 14 days, additional 45 days, and 1 August 2021 (230 days), respectively. Responses of planning to wait longer were deleted as no estimated date of receipt of vaccine could be calculated.

Exposure Assessment. Four exposures of interest were chosen as sociodemographic factors: age, gender, income, and race. Age was reduced to three categories (<18–29 years, 30–59, and 60–70+), gender was reduced to a binary variable (male, female), income was reduced to four categories (<$34,999, $35,000–$74,999, $75,000–$100,000+, Prefer not to answer), and race was made into a single 5-level categorical variable (White, Black, Asian, Hispanic, Other).

Covariates. Additional covariates considered for adjustment in multivariable models were selected from the survey related to health and vaccine hesitancy. Health-related characteristics included presence of comorbidities (yes/no), if individuals had tested positive for COVID-19 active infection before (yes/no/unsure), and BMI (continuous). Due to the large dataset and number of variables, a stepwise selection was conducted to determine the most important variables related to vaccine receipt. Vaccine-related characteristics included if individuals indicated they believed COVID-19 vaccines are safe (yes/no/not sure), if individuals were confident in the pharmaceutical company research surrounding COVID-19 vaccines (yes/no/not sure), motivations for receiving the COVID-19 vaccine (to protect a family member or close friend that is high risk for disease (yes/no); to protect themselves (yes/no); and to do their part in controlling the pandemic (yes/no)), if individuals were a frontline medical worker (yes/no), if individuals believed that doctors have the best interests of patients in mind when it comes to COVID-19 (yes/no/not sure).

Analysis. Initial Kaplan–Meier (KM) survival curves with 95% confidence intervals (CI) and corresponding log-rank tests were performed on all variables to visualize the relationships between exposures and covariates and time to receipt of vaccine. Log-rank tests were used to indicate significantly different times to receipt of vaccine among groups, and multiple comparisons tests were conducted to determine significance between more than two levels, using the Bonferroni adjustment. We used Cox proportional hazards models to estimate the association between sociodemographic factors and time to vaccine receipt in crude and adjusted models. To address potential confounding, we calculated hazard ratios (HRs) using three separate models: (1) unadjusted model with sociodemographic factors; (2) adjusted model for sociodemographic and health-related factors, and (3) adjusted model for sociodemographic, health- and vaccine-related factors. The proportional hazards assumption was assessed using log–log survival plots and tests of Schoenfeld residual variability over time. KM survival curves, log-rank, multiple comparisons, and Cox proportional hazards analyses were performed using SAS statistical software (version 14.1; StataCorp LP, College Station, TX, USA); tests of Schoenfeld residual variability over time was performed in R Studio using the “survival” package (R Studio, PBC, Boston, MA, USA).

### 2.4. Geospatial Temporal Analysis

For the geospatial temporal analysis, we performed two emerging hot spot analyses using a space-time cube, to identify spatio-temporal trends in cluster point of vaccination status or vaccine perception. For these analyses, a grid of space-time cubes is defined over the study area. The space-time cubes contain a base area corresponding to spatial distribution, and a vertical axis with time. The space-time cubes were built using horizontal dimensions of a 10 miles radius hexagonal grid fishnet and a temporal vertical axis of 2 weeks.

Each space-time cube is assigned a value by averaging the observations of variable of interest which fall within the cube and filling the neighbors with zeros. Survey responses were mapped using zip codes rather than full geocoding.

We recoded variables to perform the analysis: a vaccine perception variable was created adding up the variables evaluating respondents’ perception on vaccine safety, vaccine efficacy, confidence in the pharmaceutical research, and confidence in FDA regulations. Each variable was measured with three levels: disagree, neutral and agree with values of 0, 0.5 and 1, respectively, to obtain a proportion. Vaccination status was a dichotomous variable.

To perform the spatial temporal analysis, we used the space-time cubes created, and ran two separate emerging hotspot analysis tools using the variable for the average count data. The geospatial analysis was performed using ArcGIS^®^ Pro 2.8.3 (ESRI, Redlands, CA, USA).

### 2.5. Ethical Statement

The SC DHEC and University of Carolina institutional review boards reviewed the public health surveillance initiative protocol and determined it to be human subjects research exempt.

## 3. Results

We obtained 14,915 responses from the original survey. After data cleaning, the incomplete surveys were deleted, analyzing 12,975 surveys. Descriptive analysis from the overall sample is shown in Figure 2.

### 3.1. Principal Component Analysis

The four largest principal components explained 40% of the variance in the subpopulation of surveys completed after 7 January. The principal components and the variables within the data are correlated, and these correlations are represented by factor loadings; loadings greater than 0.40 in absolute value are generally considered indicative of a meaningful correlation [19]. Figure 3 presents the relationship between the four largest principal components and the data variables. The factor loadings are provided in Appendix A.

Principal Component 1

The first component had strong negative weights for trust in the safety of the COVID-19 vaccine (−0.62), trust of the effectiveness of the vaccine (−0.61), trust in the research process of the vaccine (−0.60), and trust in the FDA approval process of the vaccine (−0.57), along with strong negative weights for social distancing (−0.40) and wearing a mask (−0.40). Simultaneously, the first principal component had strong positive weights for receiving information on COVID-19 through health professionals (0.51), news (0.53), public health officials (0.61), government officials (0.65), television (0.46), social media (0.60), friends, family, and neighbors (0.53), and federal briefings (0.58). The first principal component also had a strong positive weight for Black race (0.40).

Principal Component 2

The second principal component contained strong positive weights for trust in the safety of the vaccine (0.40), the effectiveness of the vaccine (0.40), the research process of the vaccine (0.50), and the FDA approval process of the vaccine (0.49). Additionally, the second component contained large positive weights for trust in the government being forthcoming on information regarding the pandemic (0.43), trust in the government telling the truth about the pandemic (0.49), trust in the quality of information from the government regarding the pandemic (0.52), and trust in health care workers (0.52). This principal component also had strong negative weights for income (−0.41), being an essential worker (−0.41), and feelings of stress (−0.44), as well as a strong positive weight for receiving information from federal briefings (0.45).

Principal Component 3

The third principal component had strong positive weights for being a front-line medical worker (0.49), and strong negative weights for feeling sad, lonely, or depressed (−0.45), age (−0.61), and getting information from television (−0.46).

Principal Component 4

The fourth principal component had strong positive weights for concern of the spread of COVID-19 within the community (0.47), trust in the government being forthcoming on information regarding the pandemic (0.43), trust in the government being truthful about the pandemic (0.48), and trust in the quality of information regarding the pandemic (0.43). The fourth principal component had strong negative weights for White race (−0.49) and BMI category (−0.41). Principal component 1 is negatively associated with receiving the COVID-19 vaccine (OR 0.50, *p* < 0.001), indicating that vaccine hesitancy is associated with lack of trust in the safety, effectiveness, research process and regulatory approval process of the vaccine, non-compliance with masking and social distancing recommendations, receiving COVID-19-related information from a wide variety of sources (including less reliable sources), and Black race. Principal component 2 is positively associated with receiving the COVID-19 vaccine (OR 3.42, *p* < 0.001). This indicates that vaccine receptivity is associated with trust in the safety, effectiveness, research process and regulatory approval process of the vaccine, trust in the openness, honesty, and accuracy of information from the government surrounding COVID-19, trust in health care workers, and receiving COVID-19 information from federal briefings, while vaccine hesitancy is positively associated with income, being an essential worker, and feelings of stress. Principal components 3 and 4 were not found to have a significant relationship to vaccination status (Table 1).

### 3.2. Survival Analysis

The four exposures’ KM curves are included in Figure 4. Appendix A contains the log-rank test results and any applicable multiple comparisons test results if KM log-rank tests indicated significantly different levels from the KM survival curves—for all considered variables. All sociodemographic exposures demonstrated significant differences in time to vaccination among groups. As demonstrated in Figure 4a and Appendix A, there is a significant difference in the time to receipt of vaccine between the three age groups, where those aged 60–70+ years old received the vaccine faster than those aged 30–59, followed by those aged <18–29, in that order. Shown in Figure 4b, there is a significant difference in males’ and females’ time to vaccination, where males received the vaccine faster than females. Additionally, there were significant differences among income groups (Figure 4c): those making $75,000–$100,000+ annually received the vaccine faster than those making less than $34,999 and those making $35,000–$74,999 annually; those who preferred not to answer regarding their annual income level received the vaccine faster than those making less than $34,999 annually. Lastly, there were significant differences among the race categories as well (Figure 4d). Whites received the vaccine faster than all other race categories, and Asians received the vaccine faster than those in the ‘Other’ race category.

Shown in Appendix A, individuals indicating they have one or more comorbidities and those who never tested positive for active infection for COVID-19 received the vaccine faster than those who did not have one or more comorbidities and never tested positive for active COVID-19 infection, respectively.

Regarding thoughts on the vaccine itself, participants who indicated that they thought the COVID-19 vaccines were safe and effective received the vaccine faster than those who were unsure of these statements, and faster than those who thought the vaccines were not safe and not effective (Appendix A). Individuals who stated they were confident in the pharmaceutical research process for COVID-19 vaccines received the vaccine faster than those who were unsure, and faster than those who did not indicate they were confident in this process. Individuals who indicated that they were motivated to receive the vaccine to protect a close family member or friend who was at high risk for severe disease, protecting themselves, doing their part in controlling the pandemic, or if they were a frontline medical worker all received the vaccine faster than those who said no to any of these statements. Lastly, individuals who said that doctors do have the best interests of their patients in mind when it comes to COVID-19 received the vaccine faster than those saying they were not sure, followed by those who were unsure in that order.

Estimates from Cox proportional hazards models of risk—or receipt—of vaccine are shown in Table 2. Tests of Schoenfeld residuals indicated that the proportional hazards assumption was met for all variables in the models. Without adjustment for health- or vaccine-related covariates (Model 1), those aged 30–59 and 60–70+ had approximately 1.40 and 3.28-fold greater hazard for receipt of vaccine, respectively, compared to those aged <18–29. Additionally, individuals who made between $35,000 and $74,999, $75,000 and $100,000 annually, and those who preferred not to answer had approximately 1.10-, 1.30-, and 1.12-fold greater likelihood for receipt of vaccine, respectively compared to those who made <$34,999 annually. Lastly, Asian residents had approximately 1.20-foldgreater hazard for receipt of vaccine compared to White, while those in the ‘Other’ category had 0.66-fold the hazard for receipt of vaccine compared to White. No difference was seen in likelihood for vaccine for gender or any other race category.

Following adjustment for health-related covariates (Model 2), the same patterns were evident across the exposures of interest, except the likelihoods for vaccine receipt were strengthened. Individuals with one or more comorbidities had an increased likelihood for receipt of vaccine (HR 1.20); however, those who had never tested positive for active COVID-19 infection and those with increasing BMI had a reduced likelihood for receipt of the vaccine: hazard ratios (HRs) 0.50 and 0.99, respectively. After adjustment for both health-related and vaccine-related covariates, hazard of vaccine receipt strengthened for age groups; however, none of the income groups were significantly important for vaccine receipt. Black race had a 0.85 (95% CI 0.73, 0.98) and Other race had a 0.75 (95% CI 0.58, 0.98) likelihood for vaccine receipt compared to White race, and Asian race did not have a significant difference in likelihood for vaccine. Gender remained unsignificant. The patterns remained for the health-related covariates, albeit strengthened. In general, those who viewed the COVID-19 vaccines safe and effective and were confident in the pharmaceutical research process had an increased likelihood for receipt of vaccine, along with those who were motivated by protecting themselves, others, and controlling the pandemic. Frontline medical workers also had an increased likelihood for receipt of vaccine. However, those who were unsure if doctors had their patients’ best interests in mind had a reduced likelihood of vaccine receipt compared to those who did not believe this was true: HR 0.81 (95% CI 0.70, 0.95).

Due to the strengthened associations and inversion of likelihoods for vaccine receipt in the Black and Asian race categories, we think the fully adjusted model, Model 3, fits this dataset best.

### 3.3. Geospatial Analysis

Emerging hotspot analysis for vaccination status showed most areas had oscillating hot spots throughout the past year, indicating heterogeneity of vaccination status among the local population. These are statistically significant hot spots for the final time step that were a significant cold spot during the previous time step. Sporadic hot spots were seen surrounding the areas of oscillating hot spots close to the more populated cities in the state (Figure 5). These were intermittently statistically significant hot spots with no history of ever being statistically cold spots. There is also a consecutive hot spot, a location with consistent hot spot statistically significant in the final time step interval, prior that time step, yet less than 90% of the time steps were significant hot spots (Figure 5).

The emerging hotspot analysis for vaccine perception identified oscillating hot spots, indicating a statistically significant hot spot at the final time interval with previously cold spots for a prior time interval. Less than 90% of the time intervals were statistically significant hot spots (Figure 6). We found a cluster of sporadic cold spots indicating locations that are on and off-again cold spots. Less than 90% of the time intervals were statistically significant. This last cluster shows there is a small proportion of the population with an anti-vaccine sentiment noted in a coastal area.

## 4. Discussion

This paper assessed COVID-19 vaccination status, hesitancy, and perception through distinct, yet complementary statistical methodologies to evaluate overlapping themes in a reproducible and rigor-driven approach. All three results found that individuals are less likely to receive the COVID-19 vaccine if they do not trust in the science, research and governmental approval processes behind these vaccines. Similarly, those that thought vaccines are safe, effective and were confident with the pharmaceutical research behind the vaccines received the vaccine earlier and were more likely to receive the COVID-19 vaccine.

Principal component analysis showed two significant factors in relation to receipt of the COVID-19 vaccine: vaccine mistrust and information garnered through various forms of media (principal component 1), and trust in both the science and safety behind the COVID-19 vaccine and the government (principal component 2). Principal component 1 represented individuals with strong mistrust in the science behind the COVID-19 vaccine, and who reported fewer preventative efforts. Additionally, these individuals reported receiving their information through various forms of media. This principal component was associated with reduced odds of receiving the COVID-19 vaccine. Individuals who have a strong mistrust in the science and safety of the COVID-19 vaccine and who garner information through media are less likely to receive the COVID-19 vaccine [1,2,3]. Principal Component 2 represented individuals with strong trust in the science behind the COVID-19 vaccine and trust in the government, and who reported lower income. This component was associated with increased odds of receiving the COVID-19 vaccine. Individuals that trust the science behind the COVID-19 vaccine and the government were more likely to receive the vaccine [3,4,5].

The spatial time series showed that, overall, there was an increased tendency in vaccination rates towards the end of this study, as well as more positive vaccination perception. However, a cluster of sporadic vaccine hesitancy was seen in the coastal area north of one of the biggest cities. Moreover, the survival analysis indicated that non-White races and ethnicities received the vaccine later than White residents. Black and ‘other’ race/ethnicity had a reduced likelihood for vaccine. This gender and income disparity is most likely due to the composition of South Carolina frontline medical workers: 68% of active physicians are male and 74% are White [20]. Additionally, it is well known that there is a gender disparity among those investigating infectious diseases (related to frontline medical work): men dominate this arena, further lending evidence to this issue [21,22]. Given that frontline medical workers received the vaccine at the earliest rollout, this large number of employed (high-income), White, men likely heavily influenced this aspect of the survival analysis. Another important factor that could lend insight to the gender differences seen here is that reproductive-aged women in health care have reportedly demonstrated significantly higher rates of vaccine hesitancy, especially those trying to conceive or already pregnant [23]. These results are important to highlight as South Carolina has a relatively low vaccination rate compared to other states [17] and understanding the state’s vaccination hesitancy may help create better targeted public health decisions. Considering racial and economic disparities is important, and targeted programs providing monetary compensation for vaccination could collectively encourage more people to get the COVID-19 vaccine [24].

The PCA analysis found that lower income was part of a principal component associated with greater odds of receiving the COVID-19 vaccine, which is the opposite of what the survival analysis suggested: that those with less annual income received the vaccine significantly slower than those with higher annual income. We speculate that the discrepancy in the effect of income between the PCA and the survival analysis is due to three factors. First, income was measured categorically, limiting the precision of this variable. Furthermore, self-reported income is well known to suffer from inaccuracies due to social desirability bias, and non-response bias. Second, the dataset used for the PCA was different from that used for the survival analysis, and thus the different results may be partially explained by the different datasets used. The PCA used all surveys completed after 7 January. However, individuals who were not eligible to be vaccinated at the time of survey completion and indicated that they planned to wait longer than Fall 2021 to be vaccinated were excluded from the survival analysis, as it was impossible to estimate their time to vaccination. Therefor the dataset used for the survival analysis was biased towards individuals who intended to be vaccinated by Fall 2021. Third, the PCA and the survival analysis are answering slightly different research questions. The response in the PCA is a binary indicator of vaccination status, and the negative association between income and vaccination status found in the PCA analysis indicates that higher income people are less likely to be vaccinated overall. The response in the survival analysis is “time to vaccination”. Thus, the positive association between income and time to vaccination found in the survival analysis indicates that among this subpopulation used for the survival analysis (which is biased towards individuals who intend to vaccinate), wealthier individuals were likely to be vaccinated more quickly. This finding likely does not generalize to the whole state population, but rather suggests that among those who intend to be vaccinated, wealthier individuals get vaccinated more quickly. These findings are also likely due in part to the fact that health care workers were the first to be eligible for vaccination in South Carolina. As a large number of health care workers have a relatively high income (doctors, physicians assistants, etc.), the initially eligible population likely had a higher income than the state as a whole.

One set of our results agree with a survey performed across the United States, where they found higher-income households seem to be less likely to show vaccine hesitancy than lower-income households [25]. Differences in the geographical distribution could also be related to the inequities due to race and area of residence and could explain the vaccination uptake and hesitancy found in our results [26]. Our results also confirm previous findings linking increased likelihood of receiving the vaccine with trust in the government and trust behind the science and safety of the COVID-19 vaccine, as well as reduced likelihood of receiving the vaccine when individuals do not trust the government or the science behind the vaccine [27,28,29,30,31].

A unique aspect of the current project was the ability to utilize a state-wide year-long surveillance initiative dataset which created an opportunity for analyzing vaccine hesitancy using multiple approaches. Each of these analyses was applied to address vaccine hesitancy, and thus we believe the similarities between results are worth noting. Differences seen in the analyses, such as that seen with income, can be helpful not only for public health response and decision makers, but for study and survey design, to create appropriate measurement for those variables.

There is a growing cultural divide between childhood vaccine hesitancy and COVID-19 vaccine hesitancy. In this analysis, we assess COVID-19 vaccination hesitancy as a result of a pandemic response in South Carolina, a conservative state with high number of anti-vaccine groups [32]. While childhood vaccine hesitancy is a well-known problem in the U.S. [8], COVID-19 vaccine hesitancy emerges as a response to an unprecedented situation, and a novel vaccine technology was used in the current pandemic. In a study that conducted an assessment for COVID-19 vaccine hesitancy with more than 13,000 adults, 75.2% of the American participants seemed to be willing to be vaccinated, and yet in the U.S only 65% have reached full vaccination, and 28% the booster dose [17]. This situation seems to be influenced, in part, by the massive amount of information available online, and the misleading information shared in social media about the COVID-19 vaccines. One example article highlights anti-COVID vaccination sentiments, such as concern for vaccine adverse effects [33].

Publicly available COVID-19-related information in South Carolina was presented through multiple outlets, including billboards from the major health care system (PRISMA Health), online daily information releases on the number of hospitalizations, the Department of Health and Environmental Control online press releases, and the CDC website. Residents of the state were able to access this information freely. Future studies should explore additional methods for communicating health-related information, including integrating the health-belief model to reach at-risk groups and a broader audience [34,35]. If the population does not trust information from the government, and mistrust is built within a sector of the population, it will impact the success of the public health response. Therefore, novel communication avenues are needed to reach diverse populations with strict communities with little trust for outside members, including the federal government. Our results indicate that trust in science may be stronger in this state than trust for the government, so focusing on messages that rely on the science, rather than the government, could be beneficial [36].

The limitations for this study include the different subsets of data utilized for each analysis, indicating that caution should be used when directly comparing results. Additionally, the low overall response rate was a limitation. As this was a public health initiative operated through the state health department, no incentives were offered to participants other than receipt of their test result. This could lead to self-selection bias, thus negative perception of the COVID-19 vaccine could be less represented. Descriptive statistics indicated that the majority of participants were White (84.8–87.9%), indicating the survey did not reach a representative sample of residents of the state, as only 63.4% of the SC population is White [37]. Therefore, our sample was skewed toward racial homogeny, which could have biased results. New studies are ongoing to further explore the racial and rural health disparities of COVID-19 in our state [38]. When performing geospatial analysis, it is necessary to account for autocorrelation, and population density, to ensure values are not falsely interpreted. To account for autocorrelation, we applied false discovery rate correction to potentially reduce bias. We averaged the values to account for the underlying number of responses and prevent the detection of false hotspots in areas with higher response rates. In principal component analysis, as the principal components are linear combinations of original variables, individual independent variables become less interpretable. Additionally, the ordinal nature of the income variable could have affected the resulting association between this variable and the principal component. In addition, the survival analysis relied on each participant to tell the truth regarding whether or not they would receive the vaccine if they were one of the individuals who was not yet eligible for the vaccine. We calculated vaccine receipt time for these participants who were surveyed but not yet vaccinated, which was entirely dependent on participants following through with this. This in turn underestimated the underlying variance of the time of individuals’ vaccine receipt, which could adversely impact our p-values and confidence intervals of our survival analysis. Next, multiple demographic variables’ categories were reduced for the survival analysis in order to allow the statistical program to run, and this makes these results less comparable to the other two analyses. Lastly, while this study encompassed a representative population from a large region (entire state), the findings are not likely representative to other US states or countries.

In closing, this analysis assessed a complex issue that requires ongoing attention. Each statistical analysis is not without limitations; however, utilizing three different analyses on the same dataset strengthens our conclusions on the complex issue of vaccine hesitancy in the state of South Carolina.

## 5. Conclusions

Vaccine hesitancy regarding COVID-19 vaccines is a highly complex issue. This paper highlights the importance of the measurement of variables such as income and race to account for health disparities in future studies and potential future outbreaks requiring public vaccination and cooperation. The common themes of these analyses indicate that those with trust in the government and the COVID-19 vaccines in general are more likely to receive the vaccine, which is to be expected. Differences highlighted include the varying rates of vaccine receipt related to location in the state, income, and race. Without applying all three analyses on the dataset, some of the concerning issues regarding demographic differences among residents of the state receiving the vaccine might not have been identified. Source of information is one of the main influences on vaccination hesitancy, and work needs to be carried out on integrating health practice and behavior theory into communication to ensure reaching out to the population and attaining a successful public health response.

## Figures and Tables

**Figure 1 vaccines-10-00412-f001:**
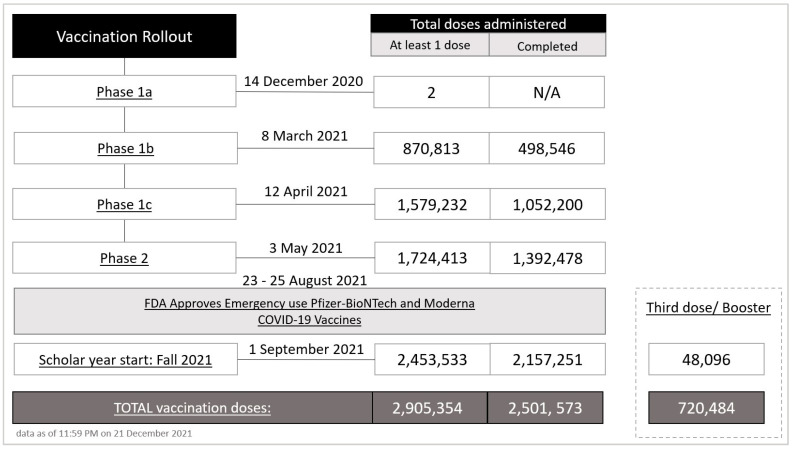
Vaccination rollout dates, corresponding pandemic phase, and the total number of doses administered including Pfizer-BioNTech and Moderna (first or completed doses), and J&J/Janssen (as complete dose): South Carolina, December 2020–December 2021. The information was obtained from the SC DHEC COVID-19 vaccine guidance and allocations data [14] and the Vaccination Dashboard [15].

**Figure 2 vaccines-10-00412-f002:**
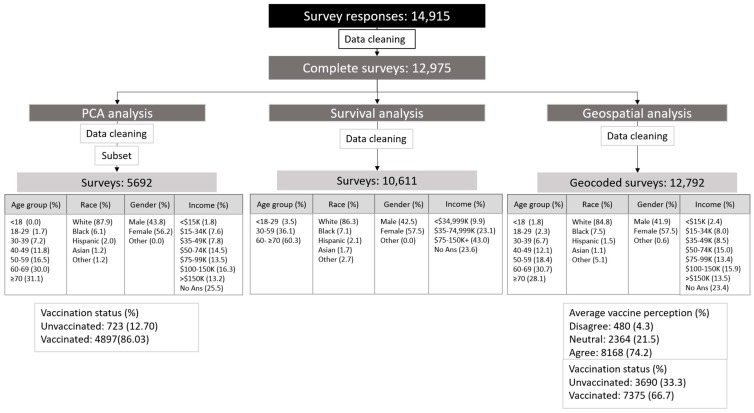
Demographics data and data process for each analysis.

**Figure 3 vaccines-10-00412-f003:**
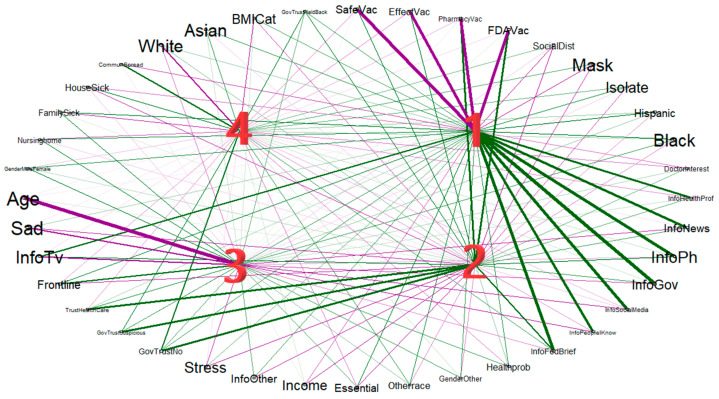
Unrotated component loadings. Variables with negative loadings are connected with a purple line, and variables with positive loadings are connected with a green line. The width of the line, along with the font size of the variable names, represents the strength of the relationship between the variables and the principal components. A list survey questions corresponding to each variable name can be found in the Appendix A.

**Figure 4 vaccines-10-00412-f004:**
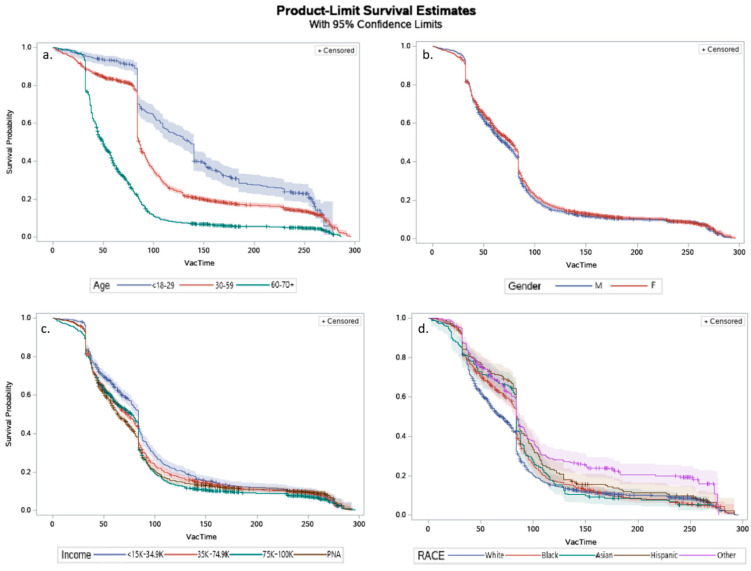
Kaplan-Meier survival curves demonstrating differences in receipt of vaccine for sociodemographic exposures: (**a**) age; (**b**) gender; (**c**) income; (**d**) race.

**Figure 5 vaccines-10-00412-f005:**
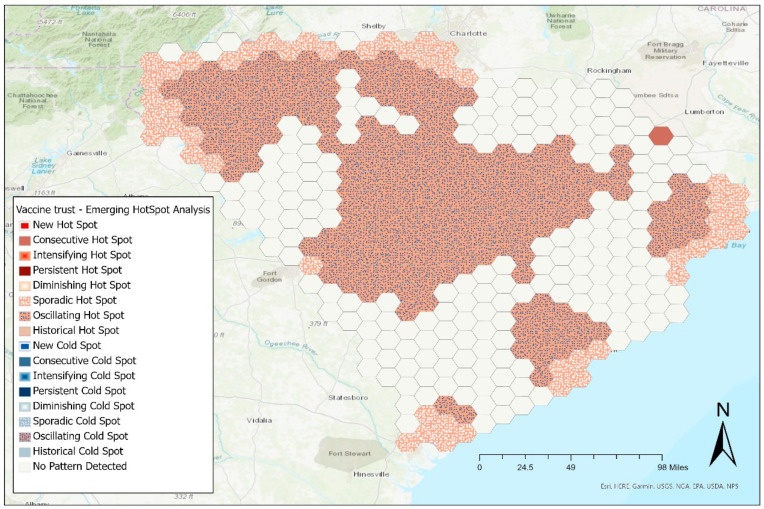
Emerging hotspot analysis of COVID-19 vaccination status among South Carolina residents, between January 2021 and October 2021.

**Figure 6 vaccines-10-00412-f006:**
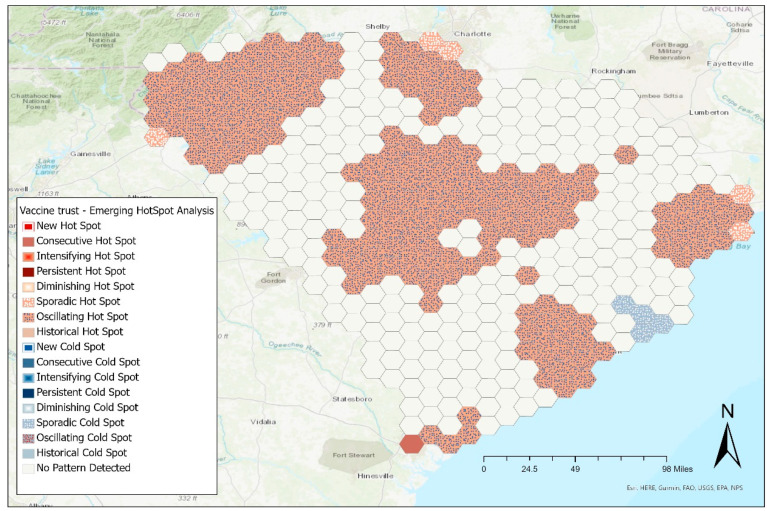
Emerging hotspot analysis of COVID-19 vaccine perception among South Carolina residents, between January 2021 and October 2021.

**Table 1 vaccines-10-00412-t001:** Logistic regression results of individual principal components and their relationship to receiving the COVID-19 vaccine.

Principal Component	Short Description	OR	95% Confidence INTERVAL	*p*-Value
PC1	Vaccine mistrust	0.50	0.45–0.58	<0.001
PC2	Vaccine and government trust	3.42	3.07–3.83	<0.001
PC3	Mixed variables of uncertain significance	1.04	0.90–1.21	0.6
PC4	Community, information, and risk factors	1.03	0.88–1.22	0.7

**Table 2 vaccines-10-00412-t002:** Hazard ratios (HRs) for vaccine receipt by cox proportional hazards model.

Variable	Model 1 * HR (95% CI)	Model 2 † HR (95%) CI	Model 3 ‡ HR (95% CI)
**Age** <18–29 30–59			
	**	**	**
	1.40 (1.24, 1.59)	1.54 (1.31, 1.81)	1.80 (1.39, 2.32)
60–70+ **Gender**	3.28 (2.90, 3.72)	3.26 (2.78, 3.83)	3.65 (2.83, 4.72)

Male Female **Annual income** <$34,999	**	**	**
	1.01 (0.97, 1.05)	1.01 (0.95, 1.06)	1.00 (0.93, 1.08)

	**	**	**
$35 K–$74,999 $75 K–$100 K+	1.10 (1.02, 1.19)	1.17 (1.05, 1.30)	0.97 (0.84, 1.13)
	1.30 (1.21, 1.40)	1.37 (1.25, 1.52)	1.10 (0.95, 1.26)
Prefer not to answer	1.120 (1.035, 1.211)	1.140 (1.027, 1.267)	1.064 (0.919, 1.233)
**Race**			
White	**	**	**
Black	1.02 (0.94, 1.11)	1.03 (0.93, 1.14)	0.85 (0.73, 0.98)
Asian	1.20 (1.03, 1.40)	1.29 (1.01, 1.63)	0.92 (0.65, 1.30)
Hispanic	0.94 (0.81, 1.08)	0.96 (0.80, 1.15)	0.81 (0.62, 1.05)
Other	0.66 (0.57, 0.75)	0.60 (0.50, 0.72)	0.75 (0.58, 0.98)
**Comorbidities**			
No		**	**
Yes		1.20 (1.13, 1.27)	1.26 (1.17, 1.37)
**Ever tested positive for COVID-19**			
No		**	**
Yes		0.49 (0.46, 0.53)	0.68 (0.62, 0.74)
**BMI**		0.99 (0.98, 1.00)	0.98 (0.97, 0.99)
**Think vaccines are safe**			
No			**
Yes			4.61 (2.52, 8.40)
Not Sure			2.49 (1.39, 4.44)
**Thinks vaccines are effective**			
No			**
Yes			1.85 (1.12, 3.06)
Not Sure			1.47 (0.90, 2.41)
**Trusts the pharmaceutical research behind vaccines**			
No			**
Yes			1.80 (1.27, 2.55)
Not Sure			1.69 (1.22, 2.35)
**Got the vaccine to protect family or friend at high risk for severe disease**			
No			**
Yes			1.09 (1.00, 1.19)
**Got the vaccine to protect self**			
No			**
Yes			2.43 (2.10, 2.82)
**Got the vaccine to help control the pandemic**			
No			**
Yes			1.30 (1.17, 1.46)
**Frontline medical worker**			
No			**
Yes			3.23 (2.79, 3.73)
**Think doctors have the best interest in patients when it comes to COVID-19.**			
No			**
Yes			0.97 (0.86, 1.09)
Not Sure			0.81 (0.70, 0.95)

* Unadjusted model. † Adjusted for comorbid, BMI, ever tested positive for COVID-19. ‡ Adjusted for Model 2 covariates and think vaccines are safe, think vaccines are effective, trusts pharmaceutical research behind vaccines, got the vaccine to protect family or friend at high risk of disease, got the vaccine to protect self, got the vaccine to help control the pandemic, is a frontliner medical worker, think doctors have the best interest in patients when it comes to COVID-19. ** Reference level.

## Data Availability

The data presented in this study are openly available in [OPENICPSR] at [doi] reference number [10.3886/E161504V1].

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
