# Peer review of "Determinants of COVID-19 Vaccinations among a State-Wide Year-Long Surveillance Initiative in a Conservative Southern State"

_vaccines, 2022, doi:10.3390/vaccines10030412_

Round 1

Reviewer 1 Report

This is an excellent analytical paper whose consequences can reach far to elucidate aspects of the real and extant problem of vaccine hesitancy as well as attitudes surrounding it. However, the manuscript needs revision for clarity before it can be accepted for publication. I have annotated the attached PDF with questions and comments. 

Author Response

Response to reviewer 1 comments

Abstract line 14: given the context on which this paper primarily focuses, I think it's highly relevant to state US statistics (case load and death) along with (or lieu of) world statistics.

Thank you for your comment. We added a context with US statistics line 13-14

Abstract Lines 14–15: "resulted in". The word choice here seems to indicate a causal relationship between "public health guidelines poorly received" and "uniqueness and gravity of this pandemic". This is contrary to what you have concluded at the end of the abstract by showing relationship between information source and vaccine receipt. I recommend you revise lines 14–15 to remove the reference to causality.

We agree the choice of wording was not appropriate, we changed the sentence in line 15 to “The uniqueness and gravity of this pandemic has been reflected in the public health guidelines poorly received…”

Abstract Line 17: This highlighted statement is so important that a simple statement of fact feels inadequate here. Is it possible to add a line providing a generalized explanation for the distinction between these two instances of vaccine hesitancy?

For simplicity of the abstract we have removed that sentence from the abstract.

Abstract line 19: I suggest using "any single..." instead of "a single..." for a closer alignment with what you say next.

Thank you for the suggestions, we agree with the comment and we made the change. Line 20

Abstract Line 21: Your statistical analyses were performed on the cleaned up 12,000+ responses, not the 14,000+ responses you received. Is it not better to state the actual number of responses you based your analyses on? (It’s okay to state both.)

Thank you for your comment. We agree is better to state the 12,975 surveys instead. We made the change, line 22

Abstract lines 25–26: This is the key result statement in the abstract. It is pivotally important, but the current word placements and punctuation are making it slightly difficult to follow the flow of concepts around which influences what. Could you please revise for clarity? Perhaps using passive voice here would help with the directionality of cause and effect.

Thank you for the suggestion. We rephrased the sentence for clarity line 26-27: “The results indicate that vaccine receipt is influenced by the information source and the population’s trust in the science and approval process behind the vaccines”

Intro lines 33–34: As with the abstract, the introductory line desperately needs more relevant US statistics to complement or supplant global statistics. Please state the overall cumulative COVID-19 cases and death numbers in the US at December 2021.

We added the statistics for the US at December 2021. Line 36-37

intro line 35: Similar to what I wrote for the abstract, this mention of "novel nature" is crying out for an explanation of said novelty of nature. Even though you have provided citations for this statement, it is important enough to merit additional clarity via a line or two explaining what is meant by "novel nature" and how exactly this novel nature has impacted preparedness and response.

Thank you for the comment. We added a sentence to explain the novel nature. Lines 38-40“The first response was to implement a generalized lockdown at the beginning of the pandemic and restricting the mobility to curb the spread of SARS-CoV-2”.

Highlighted part in lines 37–38: this is an incomplete sentence. Please revise.

We revised the sentence, and it reads: “One of these strategies was the development of guidelines to prevent the spread of COVID-19, that included promoting social distancing, handwashing, mask wearing, regular testing and vaccination”

Line 39: It is fine to cite a reference (#5) in support of "not always well received", but the gravity of this statement demands some elucidation as to the possible reasons for this ill-receipt. Could you please add 1 or 2 line(s) to that effect?

A sentence has been added for support to the mentioned statement. The paragraph lines 44-48 reads: “These guidelines are not always well received by the public, which posed a challenge for successfully preventing disease spread [5]. Meltzer et al., found that adherence to public health measures was related to worry about contracting COVID-19 [5], and in a study of Reader et al., at least 17-21% of the respondents were not likely at all to wear a mask  [6]. This previous research indicates that for successful public health responses, we must take population behaviors and disease knowledge into consideration [6]”

Line 44: Please consider both the logical and temporal sequence for this statement: Which came first, trust / knowledge or information sources? Please revise as needed.

The population’s trust and knowledge come from where they get their information from, therefore information sources will come first. For a more appropriate statement we removed “consequently” from the sentence as we are not trying to imply causality. Line 52.

Lines 45–46: If you are referring to generalized vaccines, then their goal is to prevent disease, period. The questions of "preventing severe disease" and "preventing swamping of the healthcare delivery system" are, however, relatively unique to vaccines against influenza and SARS-CoV-2, having arisen due to the observations of post-vaccination breakthrough infection and disease. Please decide which aspect, generalized or specific, you want to highlight and revise your sentence accordingly.

Thank you for the comment. The sentence is specific for the SARS-CoV-2 vaccines, we made the changes to clarify. Line 53 It reads: “Vaccines against SARS-CoV-2 are an invaluable tool to help prevent severe disease and to prevent swamping the healthcare system.”  

Line 50. Given you reference 7, please reconsider the use of "moreover" here.

We removed “moreover” of the sentence line 58.

Line 53: What about primary infections in unvaccinated persons? The mention of "reinfections" is an important consideration which merits a few words.

We agree in the importance of the primary infections in unvaccinated people, we made a few changes line 61-63. It reads” The evolution of variants has diminished the protection of currently available vaccinations, has maintained the advent of primary infections and led to a surge of reinfections among unvaccinated persons, and has brought breakthrough infections among vaccinated persons”

Figure 1 legend: please include the name(s) of the vaccines administered in phases 1a–c and 2 prior to FDA's EUA to the mRNA vaccines. (Since you mention them.)

For the doses administered data was obtained from the SC Department of Health and Environmental Control vaccine allocations data, that included information of the doses administered for Johnson&Johnson/Janssen, Moderna and Pfizer-BioNTech. We added the names in the legend lines 79-80. It reads: “Vaccination rollout dates, corresponding pandemic phase, and the total number of doses administered (including Pfizer-BioNTech and Moderna (first or completed doses), and J&J/Janssen (as complete dose):”

Line 79: In the abstract, the survey says 14,915 participants. Please clarify this discrepancy.

Thank you for the comment. WE agree that there was a discrepancy. We made the change. Line 89: “Almost 15,000 residents participated in this sampling …”

Line 89: Is it possible to add a line explaining the rationale behind changing the sampling method?

We clarified the rationale behind changing the sampling method in line 102-103, by adding the sentence: “Sampling methods were changed half-way through the surveillance initiative to allow for greater recruitment”.

Line 128: Question for authors: Do you have a sense of the proportion of your participants who had to be eliminated for this reason? I am trying to gauge if this elimination (completely necessary, I understand) led to any kind of under-representation.

We appreciate the comment. The number of participants that were eliminated for this reason are 44 (which is less than 0.5% of the total sample used for the Survival Analysis) We believe it was negligible and it should not affect the results.

Line 202: There is a discrepancy (not your fault, it's the way the data is) in the vaccinated/unvaccinated proportions between your PCA cohort and GSA cohort. Do you think this difference did or did not impact the analytical outcomes from these cohorts? How generalizable are these outcomes, is my question.

Part of our discussion considers the discrepancies between analysis, and how generalizable result are due to that reason. Because the PCA was a subset of the sample, it adds a limitation in the generalizability of the results, thus we thought that it was important to show that there are some differences between analysis that should not be ignored. We corrected the figure for a better understanding of the data cleaning process and added a few paragraphs on section 2.2.

Table 1: For better comprehension of the table 1 data, is it possible to reduce each principal component to a short text-based description and add that to an additional column?

We created a short description for each of the PCA components in the table added an additional column in table 1.

Paragraph starting Line 289: For the textual description presented in this paragraph (as well as the reference to Table S2), I am wondering if it is possible to include a tabular view of participant thoughts and their vaccine-acceptance response. I think it would greatly enhance the appreciation of your paper.

Thank you so much for your comment. To avoid overwhelming the reader, we have made the data publicly available, so that any reader interested in performing the analysis can do so on their own time.

Line 314: You probably don't need this text. :)

We removed the example title and added the correct title to the table.

Discussion. Your abstract clearly indicates the information source is an important factor. Yet, your discussion does not include any substantial input or insight on that count. Please remedy this in both your discussion and conclusion.

We added a new paragraph to address this point and expand it lines 477-490

You also state in your limitation section that your sampling may not have been representative of the state's demographics. Given that this is a response-based study, that observation itself can be a data point to spark off additional analyses.

In our forthcoming article we performed your requested analysis and described this limitation at length. That article is currently in press we just received the proofs. Pending its final publication, we will reference this paper for readers in the discussion.

I want some clear pictures to emerge from each paragraph of your description addressing the each component of vaccine hesitancy. You have already done the hard work; people, epidemiologists, policymakers as well as laypersons, whoever reads your paper should have some clear messaging to take home.

Thank you for the comment, we have revised our conclusions to make it more understandable to a broad audience. Lines 525-537

Lines 443-444: Where is your survival analysis inference? All inferences need to be clearly demarcated and stated. If need be, please break your paragraphs down to smaller chunks.

We apologize for the confusion, we have updated the wording to clarify lines 480-481

Ref 13, 14, 15 need URLs.

We added the URLs to the references.

Reference 17 is incomplete.

Completed the reference adding the URL.

Line 542, reference 28. Needs a URL.

Added the URL.

Reviewer 2 Report

Materials and Methods

Please provide more details about data cleaning.  Specifically provide more details and/or definition of internal data point validation checks.  Please also indicate how outliers were removed – for example was statistical analysis was performed and what were the criterial to establish an outlier. 

Results

Please indicate of the ~2000 samples that were not include after data cleaning, the approximate breakdown in which category they fell under. 

Similarly, please provide more information about how samples sizes were arrived to for the PCA, Survival analysis and Geospatial analyses.  Figure 2 shows another round of data cleaning - however this is not described in the methods and therefore justification and details should be included.

PCA was performed on a limited number of sample compared to the whole population. Therefore, when explaining these results, it should be noted and clarified that the four largest variables explained 40% of the variance within this subpopulation on individuals.

The authors refer to variables with negative and positive loadings (Figure 3), however the definition of that is not clear.  Although it appears that more information is described in the results sections, having a clear definition of what those terms mean or constitute in the results or materials and methods sections would be beneficial and clarifying.

When describing the results for section 3.2 (survival analysis), there is a paragraph that begins with describing the impact of thoughts on time to vaccination – it is important to refer the audience to the data in Supplementary Table S2 for all the factors discussed.  This will enhance the reader ability to find the data and assess it.

Discussion –

The authors can improve the discussion on the implications of their findings.

For example, Time to vaccination – are the results that show higher income, white males received that vaccine faster than females.  Does this reflect privilege?? Or does this reflect having access due to job classification (healthcare worker, first responder etc,), or does this access to healthcare, or some combination thereof?  Any insight on this or at least a brief discussion would positively impact the discussion.

The authors highlight a bit of a discrepancy or apparent inconsistency in the data from time to vaccination versus the spatial analysis.  Specifically as it relates to income.  Can the authors please provide some sort of speculation or insight into how these two finds co-exist?

Author Response

Response to reviewer 2 comments

Materials and Methods

Please provide more details about data cleaning.  Specifically provide more details and/or definition of internal data point validation checks.  Please also indicate how outliers were removed – for example was statistical analysis was performed and what were the criterial to establish an outlier. 

We removed survey responses that were incomplete. We cleaned data for values that were impossible for today’s date, vaccination date, BMI, and weight. We did not test for any outliers. We manually removed the values that had an impossible vaccination date and today’s date (before the start of the survey, before the first rollout of the vaccine in South Carolina) and values for weight that were above 700lbs or values for BMI above 300. All zip codes not in SC were deleted.

Results

Please indicate of the ~2000 samples that were not include after data cleaning, the approximate breakdown in which category they fell under. 

Thank you for your comment. In the first data cleaning we removed those surveys that were incomplete and had mostly missing answers (redcap provides a column that says if the survey is complete/incomplete) or were performed by the participant as a first attempt but were corrected in a second attempt. Line 213 We mention the surveys were deleted due to incompleteness we did not think there was any need for more clarification.

Similarly, please provide more information about how samples sizes were arrived to for the PCA, Survival analysis and Geospatial analyses.  Figure 2 shows another round of data cleaning - however this is not described in the methods and therefore justification and details should be included.

Participants who completed the survey prior to January 7th were not asked key questions about vaccine/government trust that we wished to include in the PCA. As a result, the PCA analysis was performed only on the 5,692 survey responses collected after January 7th.

PCA was performed on a limited number of sample compared to the whole population. Therefore, when explaining these results, it should be noted and clarified that the four largest variables explained 40% of the variance within this subpopulation on individuals.

Thank you for point this out. We have added this to Section 3.1.

The authors refer to variables with negative and positive loadings (Figure 3), however the definition of that is not clear.  Although it appears that more information is described in the results sections, having a clear definition of what those terms mean or constitute in the results or materials and methods sections would be beneficial and clarifying.

Each principal component is a linear combination of the original variables. The loadings associated with each principal component are the coefficients involved in this linear combination. A positive (negative) loading indicates a positive (negative) association between the principal component and the variable in question, with larger magnitude loadings corresponding to stronger relationships. We have added this clarification Section 2.1.

When describing the results for section 3.2 (survival analysis), there is a paragraph that begins with describing the impact of thoughts on time to vaccination – it is important to refer the audience to the data in Supplementary Table S2 for all the factors discussed.  This will enhance the reader ability to find the data and assess it.

Thank you for the comment. We addressed this by adding a reference to the Table S2 from the Supplementary Material. (line 299)

Discussion –

 The authors can improve the discussion on the implications of their findings.

 We addressed this point by adding additional paragraphs in the discussion.

For example, Time to vaccination – are the results that show higher income, white males received that vaccine faster than females.  Does this reflect privilege?? Or does this reflect having access due to job classification (healthcare worker, first responder etc,), or does this access to healthcare, or some combination thereof?  Any insight on this or at least a brief discussion would positively impact the discussion.

We added additional sentences to the discussion addressing this point. Lines 410-419

The authors highlight a bit of a discrepancy or apparent inconsistency in the data from time to vaccination versus the spatial analysis.  Specifically as it relates to income.  Can the authors please provide some sort of speculation or insight into how these two finds co-exist?

Vaccine hesitancy is complex and multifactorial. The purpose of this study was to use a variety of statistical techniques to understand different aspects of vaccine hesitancy and to determine which trends are robust to the analytical method in question. We speculate that the discrepancy in the effect of income between the PCA and the survival analysis is due to three factors. First, income was measured categorically, limiting the precision of this variable. Furthermore, self-reported income is well-known to suffer from inaccuracies due to social desirability bias, and non-response bias. Second, the dataset used for the PCA was different from that used for the survival analysis, and thus the different results may be partially explained by the different datasets used. Third, the PCA and the survival analysis are answering slightly different research questions. The response in the PCA is a binary indicator of vaccination status, and the negative association between income and vaccination status found in the PCA analysis indicates that higher income people are less likely to be vaccinated overall. The response in the survival analysis is time to vaccination. Notably, individuals who were not eligible to be vaccinated at the time of survey completion and indicated that they planned to wait longer than Fall 2021 to be vaccinated were excluded from the survival analysis, as it was impossible to estimate their time to vaccination. Therefor the dataset used for the survival analysis was biased towards individuals who intended to be vaccinated by Fall 2021. Thus, the positive association between income and time to vaccination found in the survival analysis indicates that among this subpopulation with greater pro-vaccine sentiment, wealthier individuals were likely to be vaccinated more quickly. This finding likely does not generalize to the whole state population. These findings are also likely due in part to the fact that healthcare workers were the first to be eligible for vaccination in South Carolina. As a large number of healthcare workers are relatively high income (doctors, physicians assistants, etc.), the initially eligible population was likely higher income than the state as a whole and also had the shortest time to vaccination on average. We have added a paragraph to this effect to the discussion.

Reviewer 3 Report

Review of ‘Determinants of COVID-19 vaccinations among a state-wide year-long surveillance initiative in a conservative Southern state‘ by  Gual-Gonzalez  et al.

This study used 14,915 surveys, for a one-year period (Fall 2020 to Fall 2021) from state-wide surveillance initiative to identify themes of COVID vaccine hesitancy and vaccine trust in a US state. It performed three different statistical analyses to evaluate vaccine hesitancy: principal component analysis, survival analysis and spatial-time series analysis. This is prepared so that policymakers can use it to improve vaccine interventions. I recommend a major revision.

Main point:

  1. What are the main new findings not very clear. It is better to draw some differences between the general anti-vaccine attitude to that from COVID’s anti-vaccine attitude.
  2. Line 370: You mentioned, “All three results found that individuals are less likely to receive the COVID-19 vaccine if they do not trust in the science research and governmental approval processes behind the vaccines.” This statement is related to Anti-vaccination. There is a clear distinction between ‘ Anti vaccination’ and  ‘Anti COVID-19 vaccination’. Those who are ‘Anti COVID-19 vaccination’ almost all took other vaccines and hence definitely do not belong to the group of ‘ Anti vaccination’. Throughout use the term ‘Anti COVID vaccination’ rather than ‘Anti vaccination’.
  3. Though you mentioned in line 15 ‘COVID-19 vaccine hesitancy is distinct from childhood vaccination hesitancy, and this psychosocial issue is highly complex.’ Some elaboration of this area in the introduction with few references will be useful. Also, discuss with reference that for COVID-19 vaccine, immunity wanes in four months’ time and Israel started 4th Dose. After Israel started 4th dose the deaths in Israel became all-time high. How those could be related to your statement in Table S1: the first two rows ‘Vaccine are safe’ and ‘vaccine are effective’. The first two principal components are shown in bold.
  4. In line 24: You indicated the purpose of this study is to “identify overlapping themes of vaccine hesitancy and vaccine trust.” Some root causes of COVID vaccine hesitancy were not at all discussed which should be analysed from a critical viewpoint. To address those important issues constructively could be a step forward to improve vaccine intake. People willingly take small pax vaccine, MMR and many other vaccines, but then why so much resistance for the COVID vaccine? What are such strong motivations for a certain group of people who are not hesitant to lose their jobs and livelihood? Those issues need to be discussed too to have balanced analyses. Adverse reactions to the COVID-19 vaccine need some mention which is much higher than any existing approved vaccine. All those related important questionaries for ‘anti-COVID vaccination’ are omitted.

A recent review nicely discussed the scientific basis of many adverse reactions of the COVID-19 vaccine and that paper is worth mentioning.

Reference:

Seneff, S., & Nigh, G. (2021). Worse Than the Disease? Reviewing Some Possible Unintended Consequences of the mRNA Vaccines Against COVID-19. International Journal of Vaccine Theory, Practice, and Research, 2(1), 38–79.

        5. Fig. 2: For Principal Component Analyses (PCA) there is a total of 5,692 cases, out of which only 723 were unvaccinated. How robust could be your conclusion?

        6. Table S3, Relating to point 29 on vaccinating children starting from 6 months: parents should be clearly made aware of a few points: a) death rate due to COVID among children in various age groups is practically negligible. b) even after vaccination, any children can transmit the disease and even die from it. c) effect of vaccination dissipates in 4 months’ time and how many vaccination doses the parents will be ok with.  d) parents should be made aware of adverse reactions if any, (all possible medium term and long term) to the vaccine. Though the chances are extremely rare, still there are precedents that healthy people including children died within one to 2 days after vaccination. Most deaths after vaccination occurred within the first week and make those points clear to parents. In most cases, authorities mentioned reasons are unclear and how to gain confidence in parents and are people convinced with explanations.   

Make a few questionaries on those areas too in that Table. If difficult to gather responses to those questionaries, at least discuss those points in the text.

Minor Comments.

  1. Line 50: “Moreover, this hesitancy has resulted in relatively low vaccination rates, contributing to new SARS-CoV-2 variants (i.e., B.1.1.7, delta, 51 omicron, etc.).”

          Vaccine hesitancy did not contribute to new variants. Correct that statement.

        2. Line 52: “The evolution of variants has diminished the protection of currently available vaccinations and has led to a surge of reinfections among unvaccinated persons and breakthrough infections among vaccinated persons.” It is similar to Flu as new variants always emerge despite several new vaccines being developed for Flu from time to time. You mention that point with relevant references for Flu.

         3. Line 83: “serves as a representative population for conservative anti-vaccination residents nationally.” The term anti-vaccination resident is misleading here as there is no reference or proof whether they are against all vaccination. The more appropriate term could be ‘Anti COVID-19 vaccination’

        4. Line 84:  again ‘Anti COVID-19 vaccination’ will be more appropriate.

        5. Line 89: the sentence is not complete.

        6. Line 22: use ‘COVID vaccine hesitancy’

        7. Line 370: In the Discussion part you mentioned “individuals are less likely to receive the COVID-19 vaccine if they do not trust in the science research and governmental approval processes behind the vaccines. Similarly, those that thought vaccines are safe, effective and were confident with the pharmaceutical research behind the vaccines, received the vaccine earlier were more likely to receive the COVID-19 vaccine.”

Those statements are true for the antivaccine agenda in general. For the COVID vaccine, many doctors, many scientists and many highly educated people did not take the vaccine and they have high regard for science.

       8. Supplementary part, Table S3- points 33, 34: how essential are those? Mention those are included for BMI index calculation and to check whether those people are obese or not.

       9. Line 117: “Variables that have absolute loadings values of 0.40 or greater are considered to have a moderate contribution and will be highlighted in our analysis.”

Supplementary part: Table S1- Self isolation is one most important part, but why not bold? If any ideas, discuss briefly.

Similar to the statement. ‘Thinks doctors have the best interests in patients when it comes to COVID-19’. I expected it is a very strong statement as doctors are trying hard throughout the pandemic. Why not a single bold in this statement?

10. Table S3: 11A ii you mentioned vaccine manufacturer. You might like to do some analyses/ conclusions based on vaccine manufacturer.

Author Response

Response to reviewer 3 comments

Review of ‘Determinants of COVID-19 vaccinations among a state-wide year-long surveillance initiative in a conservative Southern state‘ by Gual-Gonzalez et al.

This study used 14,915 surveys, for a one-year period (Fall 2020 to Fall 2021) from state-wide surveillance initiative to identify themes of COVID vaccine hesitancy and vaccine trust in a US state. It performed three different statistical analyses to evaluate vaccine hesitancy: principal component analysis, survival analysis and spatial-time series analysis. This is prepared so that policymakers can use it to improve vaccine interventions. I recommend a major revision.

Main point:

What are the main new findings not very clear. It is better to draw some differences between the general anti-vaccine attitude to that from COVID’s anti-vaccine attitude.

Thank you for your comment. We have revised the conclusion section to more clearly reflect the overall pertinent findings. For the reviewer’s clarification, the manuscript was focused on vaccine hesitancy towards Covid-19 and the possible source of this hesitancy. We found in our results that information source and health disparities play a role in vaccination receipt and hesitancy and discussed the main findings in the discussion. We summarized the main take home message at the end in the conclusion paragraph. Vaccine hesitancy is complex and a more exhausting evaluation can be performed with further analyses.

Line 370: You mentioned, “All three results found that individuals are less likely to receive the COVID-19 vaccine if they do not trust in the science research and governmental approval processes behind the vaccines.” This statement is related to Anti-vaccination. There is a clear distinction between ‘ Anti vaccination’ and ‘Anti COVID-19 vaccination’. Those who are ‘Anti COVID-19 vaccination’ almost all took other vaccines and hence definitely do not belong to the group of ‘ Anti vaccination’. Throughout use the term ‘Anti COVID vaccination’ rather than ‘Anti vaccination’.

Thank you for your comment. All the questionnaire was focused only on the COVID-19 vaccines and there was one answer that was “I do not want to receive the vaccine due to personal beliefs” which was directed towards general anti-vaccines and not COVID-19 anti vaccines only. To clarify the statement we pointed out that the three results were for those that trust the COVID-19 vaccination research process and governmental approval in line 390.

Though you mentioned in line 15 ‘COVID-19 vaccine hesitancy is distinct from childhood vaccination hesitancy, and this psychosocial issue is highly complex.’ Some elaboration of this area in the introduction with few references will be useful. Also, discuss with reference that for COVID-19 vaccine, immunity wanes in four months’ time and Israel started 4th Dose. After Israel started 4th dose the deaths in Israel became all-time high. How those could be related to your statement in Table S1: the first two rows ‘Vaccine are safe’ and ‘vaccine are effective’. The first two principal components are shown in bold.

Thank you for your comment. The line was removed from the abstract to avoid confusion. The paper is mainly focused on COVID-19 vaccines only, and the sample is from a particular conservative population in the United States, therefore it is not generalizable to other populations. There are multiple factors that play in the COVID-19 vaccine hesitancy and they are related to political situation within each country, that is why we made a discussion point in lines 417-427 and 477-490 on the source of information, focused in the state of South Carolina.

In line 24: You indicated the purpose of this study is to “identify overlapping themes of vaccine hesitancy and vaccine trust.” Some root causes of COVID vaccine hesitancy were not at all discussed which should be analysed from a critical viewpoint. To address those important issues constructively could be a step forward to improve vaccine intake. People willingly take small pax vaccine, MMR and many other vaccines, but then why so much resistance for the COVID vaccine? What are such strong motivations for a certain group of people who are not hesitant to lose their jobs and livelihood? Those issues need to be discussed too to have balanced analyses. Adverse reactions to the COVID-19 vaccine need some mention which is much higher than any existing approved vaccine. All those related important questionaries for ‘anti-COVID vaccination’ are omitted.

A recent review nicely discussed the scientific basis of many adverse reactions of the COVID- 19 vaccine and that paper is worth mentioning.

Reference:

Seneff, S., & Nigh, G. (2021). Worse Than the Disease? Reviewing Some Possible Unintended Consequences of the mRNA Vaccines Against COVID-19. International Journal of Vaccine Theory, Practice, and Research, 2(1), 38–79.

Thank you for the recommendation; however, we feel uncomfortable citing this reference giving its contradiction with our field of Public Health.

In the United States, vaccine hesitancy is greater than in other countries, specially among conservative population. Our analysis was focused on the population in South Carolina, which has reached 62.9% of vaccines (one dose) and 54% of fully vaccinated individuals as of 24/02/2022. The state is one of the lowest vaccination rates in the country and this paper aims to address that.  We added a new paragraph addressing this point in lines 478-487.

Fig. 2: For Principal Component Analyses (PCA) there is a total of 5,692 cases, out of which  only 723 were unvaccinated. How robust could be your conclusion?

The analysis is robust but is not generalizable to other populations. In the discussion we address the limitation, and we highlight how different analysis for a shared issue may lead to different interpretations of the results.  

Table S3, Relating to point 29 on vaccinating children starting from 6 months: parents should be clearly made aware of a few points: a) death rate due to COVID among children in various age groups is practically negligible. b) even after vaccination, any children can transmit the disease and even die from it. c) effect of vaccination dissipates in 4 months’ time and how many vaccination doses the parents will be ok with. d) parents should be made aware of adverse reactions if any, (all possible medium term and long term) to the vaccine. Though the chances are extremely rare, still there are precedents that healthy people including children died within one to 2 days after vaccination. Most deaths after vaccination occurred within the first week and make those points clear to parents. In most cases, authorities mentioned reasons are unclear and how to gain confidence in parents and are people convinced with explanations.

Make a few questionaries on those areas too in that Table. If difficult to gather responses to those questionaries, at least discuss those points in the text.

Thank you for your comment. We appreciate the insights, and we will consider the additional evaluation on further studies. Our questionnaire did not address those points and we consider the discussion is extensive as it is, thus adding another paragraph would overwhelm the reader. Vaccination hesitancy is a complex topic and hopefully future meta-analysis can have a broad discussion on all the points that affect vaccine reluctance among the population.

Minor Comments.

Line 50: “Moreover, this hesitancy has resulted in relatively low vaccination rates, contributing to new SARS-CoV-2 variants (i.e., B.1.1.7, delta, 51 omicron, etc.).” Vaccine hesitancy did not contribute to new variants. Correct that statement.

Thank you for your comment. We agree that the low vaccination rates did not contribute to the appearance of the variance but contributed to a higher burden of disease among those non-vaccinated. The statement has been corrected to “This hesitancy has resulted in relatively low vaccination rates, contributing to increased burden of the new SARS-CoV-2 variants (i.e., B.1.1.7, delta, omicron, etc.)” (line 59)

Line 52: “The evolution of variants has diminished the protection of currently available vaccinations and has led to a surge of reinfections among unvaccinated persons and breakthrough infections among vaccinated persons.” It is similar to Flu as new variants always emerge despite several new vaccines being developed for Flu from time to time. You mention that point with relevant references for Flu.

Line 83: “serves as a representative population for conservative anti-vaccination residents nationally.” The term anti-vaccination resident is misleading here as there is no reference or proof whether they are against all vaccination. The more appropriate term could be ‘Anti COVID-19 vaccination’

Thank you for your comment. We made the clarification.

Line 84: again ‘Anti COVID-19 vaccination’ will be more appropriate.

We made the change.

Line 89: the sentence is not complete.

We made the appropriate change. It reads in line 102.

Line 22: use ‘COVID vaccine hesitancy’

We made the change

Line 370: In the Discussion part you mentioned “individuals are less likely to receive the COVID-19 vaccine if they do not trust in the science research and governmental approval processes behind the vaccines. Similarly, those that thought vaccines are safe, effective and were confident with the pharmaceutical research behind the vaccines, received the vaccine earlier were more likely to receive the COVID-19 vaccine.”

Those statements are true for the antivaccine agenda in general. For the COVID vaccine, many doctors, many scientists and many highly educated people did not take the vaccine and they have high regard for science.

Supplementary part, Table S3- points 33, 34: how essential are those? Mention those are included for BMI index calculation and to check whether those people are obese or not.

The table S3 is the survey that was sent to participants. These variables were used to calculate the BMI index and it was included in the PCA.

Line 117: “Variables that have absolute loadings values of 0.40 or greater are considered to have a moderate contribution and will be highlighted in our analysis.”

Supplementary part: Table S1- Self isolation is one most important part, but why not bold? If any ideas, discuss briefly.

Similar to the statement. ‘Thinks doctors have the best interests in patients when it comes to COVID-19’. I expected it is a very strong statement as doctors are trying hard throughout the pandemic. Why not a single bold in this statement?

The bold values are for those loadings that are greater than 0.40 as stated in the methods. If these variables did not result in an influence for the PCA analysis they were not bolded. The results of the analysis were

Table S3: 11A ii you mentioned vaccine manufacturer. You might like to do some analyses/ conclusions based on vaccine manufacturer.

Thank you for your comment. We did ask for manufacturer; however, we had low response rate as this information was an open-ended text box. Therefore, the subsequent analysis had low power and was not statistically meaningful.   

Round 2

Reviewer 1 Report

This version is MUCH improved. I am glad the authors have polished the manuscript by providing clarity in places. The additional words and paragraphs are welcome; they enhance this manuscript. There are some minor changes still required, including in the references. I have indicated them in the annotated PDF (attached). But seriously, congratulations to the authors on a job well done!

Reviewer 3 Report

The second review of ‘Determinants of COVID-19 vaccinations among a state-wide year-long surveillance initiative in a conservative Southern state‘ by  Gual-Gonzalez et al.

Some comments were attended from my last review and it is improved. However, many critical analyses part, I mentioned that could improve the quality of this paper were not addressed. Hence, I recommend a major revision.

Main points:

1. In Conclusion, you mentioned in the first line ‘ Vaccine hesitancy is a highly complex issue.’ It looks like you are addressing vaccine hesitancy here, but it is not the case. This study only focuses on COVID vaccine hesitancy. People who did not take the COVID vaccine, almost all took the childhood vaccine or some other approved vaccine. Modify conclusions accordingly.

          Also, in the earlier abstract, there was a very relevant line “COVID-19  vaccine hesitancy is distinct from childhood vaccination hesitancy….” You skipped that important line in the new abstract and may include it.

2. Line 24: You mentioned, “The results indicate that vaccine receipt is influenced by the information source and the population’s trust in the science and approval process behind the vaccines”- what about the outcome of vaccines? Will not the public observe/scrutinize the results of vaccination among the mass population? Many records of government data are now open source and people can easily check those results. If vaccine effectiveness is meeting the standard of general expectation of the public, the vaccine intakes are likely to increase. On the other hand, if people notice vaccines are not performing as was claimed initially, if there are biased media coverages and there is a lack of transparency in communicating many adverse reactions, people are likely to lose confidence. Addressing those areas and building confidence among people are likely to increase COVID vaccine intake. It is now already 2 years of the pandemic and it is more than one year since the mass vaccination started. Many countries have very high vaccination rates and people are likely to follow observed data and results in those countries. Are the high vaccinated countries performing better than the countries with low vaccination rates? Add little discussions from those angles too. Those are some areas among others that contribute to COVID vaccine hesitancy too.

3.  In your reply to my comment you mentioned, “ Vaccine hesitancy is complex and a more exhausting evaluation can be performed with further analyses.” Without obvious critical analyses on COVID vaccine hesitancy, this paper can not add much value.

         We need to think about whether the designs of the questionnaires are formulated to reach expected, predefined results or not. Though in the revision you included some limitations of this study, many relevant critical questionnaires are missing. At least those areas need to be discussed at some point in the text; otherwise, the conclusion adds little value addition to our existing knowledge.  

4.  Your answer to my comment (2nd last reply): “The bold values are for those loadings that are greater than 0.40 as stated in the methods. If these variables did not result in an influence for the PCA analysis they were not bolded. The results of the analysis were” – you did not complete this sentence. Hence there is no reply to that comment.

Minor points:

1.  You added a new line (Line 45): “in a study by Reader et al., at least 17-21% of the respondents were not likely at all to wear a mask”- but mask-wearing was mandatory.

2.  Line 56: “This hesitancy has resulted in relatively low vaccination rates, contributing to increased burden of the new SARS-CoV-2 variants (i.e., B.1.1.7, delta, omicron, etc.).” Many countries have very high vaccination rates though showed an unprecedented rise in cases. You may mention the situation of Gibraltar where the vaccination rate was achieved 100% long ago but Gibraltar showed an unprecedented rise in cases this winter.

3.  Line 59: Evaluation of new variants will always happen and the situation of Flu may be referred to here. In spite of many new Flu vaccines, new variants always emerge.  

4.  Good to see that you now included the ‘short descriptions’ column in Table 1 and it is clearer now.

5.  In conclusion, you mentioned, “Source of information is one of the main influences on vaccination hesitancy, and work needs to be done on integrating health practice and behavior theory into communication, to ensure reaching out to the population and attain a successful public health response.” Working in the area of improved vaccine effectiveness, reducing adverse reactions, transparency in disseminating balanced view are some of the important areas that can reduce vaccine hesitancy and needs to be mentioned too.

6.  Line 476: You mentioned “A study that conducted an assessment for COVID-19 vaccine hesitancy with more than 13,000 adults, 75.2% of the American participants seemed to be willing to be vaccinated, and yet in the U.S only 65% have reached full vaccination……... This situation seems to be influenced by the massive amount of information available online and the misleading information shared in social about the COVID-19 vaccines.” How do you expect 13,000 participants chosen for the assessment are a true representation of US population? Those percentages are likely to be varied for various reasons: if different locations are chosen; if the number of participants is much higher or lower than 13,000; if the sample population is dominated by different ethnic groups, age groups, income groups etc.. Moreover, following vaccine success stories and observing published government records, people are also likely to change their minds from time to time. Thus, the reduced percentage here (from 75.2% to 65%) can not be solely linked with social media misinformation. Caution should be there in making similar statements.    

Author Response

Response to Reviewer 3 comments

 The second review of ‘Determinants of COVID-19 vaccinations among a state-wide year-long surveillance initiative in a conservative Southern state‘ by Gual-Gonzalez et al.

Some comments were attended from my last review and it is improved. However, many critical analyses part, I mentioned that could improve the quality of this paper were not addressed. Hence, I recommend a major revision.

Main points:

  1. In Conclusion, you mentioned in the first line ‘ Vaccine hesitancy is a highly complex issue.’ It looks like you are addressing vaccine hesitancy here, but it is not the case. This study only focuses on COVID vaccine hesitancy. People who did not take the COVID vaccine, almost all took the childhood vaccine or some other approved vaccines. Modify conclusions accordingly.

Thank you for your comment. We have corrected this sentence to ensure that COVID-19 vaccine hesitancy is clearly described.  (line 538)

Also, in the earlier abstract, there was a very relevant line “COVID-19 vaccine hesitancy is distinct from childhood vaccination hesitancy….” You skipped that important line in the new abstract and may include it.

Thank you for your comment. The two other reviewers requested this line be removed from the abstract, where the intricacies of this comment cannot be fully addressed. It is still addressed in the discussion, as you requested.

  1. Line 24: You mentioned, “The results indicate that vaccine receipt is influenced by the information source and the population’s trust in the science and approval process behind the vaccines”- what about the outcome of vaccines? Will not the public observe/scrutinize the results of vaccination among the mass population? Many records of government data are now open source and people can easily check those results. If vaccine effectiveness is meeting the standard of general expectation of the public, the vaccine intakes are likely to increase. On the other hand, if people notice vaccines are not performing as was claimed initially, if there are biased media coverages and there is a lack of transparency in communicating many adverse reactions, people are likely to lose confidence. Addressing those areas and building confidence among people are likely to increase COVID vaccine intake. It is now already 2 years of the pandemic and it is more than one year since the mass vaccination started. Many countries have very high vaccination rates and people are likely to follow observed data and results in those countries. Are the high vaccinated countries performing better than the countries with low vaccination rates? Add little discussions from those angles too. Those are some areas among others that contribute to COVID vaccine hesitancy too.

Thank you for your comment. We appreciate this is an important variable in the overall COVID-19 vaccine hesitancy dilemma. Unfortunately, we did not include questions related to this aspect, and cannot make any substantial references to this aspect in the abstract. To satisfy the reviewer, we have added a sentence in the discussion that addresses the point of perceived vaccine ineffectiveness.   (line 480-484)

  1. In your reply to my comment you mentioned, “ Vaccine hesitancy is complex and a more exhausting evaluation can be performed with further analyses.” Without obvious critical analyses on COVID vaccine hesitancy, this paper can not add much value.

We need to think about whether the designs of the questionnaires are formulated to reach expected, predefined results or not. Though in the revision you included some limitations of this study, many relevant critical questionnaires are missing. At least those areas need to be discussed at some point in the text; otherwise, the conclusion adds little value addition to our existing knowledge.

We as the authors are sad to hear you feel the article is scientifically unvaluable. We feel this article addresses an important knowledge gap using a rigorous three-statistical approach. South Carolina has one of the fewest vaccinated populations in the United States, and this state-wide representative survey and associated surveillance initiative provide keen insight as to the motivations or dismotivations for COVID-19 vaccination in our state. The validity of the survey design and questions are routed in our collaboration with the Patient Engagement Studio (https://sc.edu/about/centers_institutes/patient_engagement_studio/index.php), which included a representative panel of lay audience members from a diverse background. This group spent hours providing their perspectives and recommendations to enhance the survey portion, which was used in the current analysis. The fact that we did work with ‘real people’ when designing the survey, the ability to leverage a state-wide surveillance initiative, and the usage of a three-part statistical approach, in our opinion adds to the scientific rigor and value of our article.

  1. In your answer to my comment (2nd last reply): “The bold values are for those loadings that are greater than 0.40 as stated in the methods. If these variables did not result in an influence for the PCA analysis they were not bolded. The results of the analysis were” – you did not complete the sentence. Hence there is no reply to that comment.

We apologize for the uncomplete response. This has been corrected.

Minor points:

  1. You added a new line (Line 45): “in a study by Reader et al., at least 17-21% of the respondents were not likely at all to wear a mask”- but mask-wearing was mandatory.

The study was performed evaluating the likelihood of mask wearing among the population from a survey performed to 378, 207 respondents from all 50 states in July 2020. At that time Mask mandates were not the same in all states. We did cite the article as the population’s responses were such.

  1. Line 56: “This hesitancy has resulted in relatively low vaccination rates, contributing to increased burden of the new SARS-CoV-2 variants (i.e., B.1.1.7, delta, omicron, etc.).” Many countries have very high vaccination rates though showed an unprecedented rise in cases. You may mention the situation of Gibraltar where the vaccination rate was achieved 100% long ago but Gibraltar showed an unprecedented rise in cases this winter.

Thank you for your comment. It is true that vaccinated persons can be infected with the omicron variant; however, the rise of recent variants occurred in vulnerable unvaccinated persons and populations. https://time.com/6124436/omicron-vaccine-inequity/

  1. Line 59: Evaluation of new variants will always happen and the situation of Flu may be referred to here. In spite of many new Flu vaccines, new variants always emerge.

Thank you for the recommendation. Flu variants emerge due to the poor flu vaccination rate. To keep the article streamlined and clear for the reader, we will refer to COVID-19 vaccine hesitancy, as this is the main article topic. 

  1. Good to see that you now included the ‘short descriptions’ column in Table 1 and it is clearer now.

We are glad this change satisfied your request.

  1. In conclusion, you mentioned, “Source of information is one of the main influences on vaccination hesitancy, and work needs to be done on integrating health practice and behavior theory into communication, to ensure reaching out to the population and attain a successful public health response.” Working in the area of improved vaccine effectiveness, reducing adverse reactions, transparency in disseminating balanced view are some of the important areas those can reduce vaccine hesitancy and needs to be mentioned too.

Thank you for your comment. The discussion is regarding the source of information the population in South Carolina relied most. We discussed the main points and what would be a good way to address the lack of trust of information. This includes any scientific information related with the pandemic and the vaccines and we did not want to engage In details to not overwhelm the reader.

  1. Line 476: You mentioned “A study that conducted an assessment for COVID-19 vaccine hesitancy with more than 13,000 adults, 75.2% of the American participants seemed to be willing to be vaccinated, and yet in the U.S only 65% have reached full vaccination……... This situation seems to be influenced by the massive amount of information available online and the misleading information shared in social about the COVID-19 vaccines.” How do you expect 13,000 participants chosen for the assessment are a true representation of US population? Those percentages are likely to be varied for various reasons: if different locations are chosen; if the number of participants is much higher or lower than 13,000; if the sample population is dominated by different ethnic groups, age groups, income groups etc.. Moreover, following vaccine success stories and observing published government records, people are also likely to change their minds from time to time. Thus, the reduced percentage here (from 75.2% to 65%) can not be solely linked with social media misinformation. Caution should be there in making similar statements.

Thank you for the comment. We have revised this sentence to read: This situation seems to be influenced, in part, by the massive amount of information available online and the misleading information shared in social media about the COVID-19 vaccines.

Round 3

Reviewer 3 Report

The third review of ‘Determinants of COVID-19 vaccinations among a state-wide year-long surveillance initiative in a conservative Southern state‘ by  Gual-Gonzalez  et al.

Some comments were attended though few critical analyses part, I mentioned that could improve the quality of this paper were not addressed.

  1. Relating to Covid vaccine hesitancy, I mentioned some points in the previous reviews, which contribute to vaccine hesitancy too. Those are not discussed.                                                                                         a) Social misinformation vs careful notice and compare observed available published government/ reputed data.                                     b) Vaccinating the whole global population every 4 months interval and the feasibility.                                                                                   c) Israel started the 4th dose and death reached an all-time high.       d) Gibraltar 100% vaccination for a long time and cases this winter were at an all-time high.                                                                         e) Vaccinating children even from age 6 months onwards (as you mentioned) and its practical implications.                                                 f) Performances of Flu vaccination, the emergence of many new variants.                                                                                                      g) Various adverse effects of vaccination and scientific basis are discussed in this nice review

                      Reference:

                      Seneff, S., & Nigh, G. (2021). Worse Than the Disease? Reviewing Some Possible Unintended Consequences of the mRNA Vaccines Against COVID-19. International Journal of Vaccine Theory, Practice, and Research, 2(1), 38–79.

You mentioned that in some cases you feel uncomfortable as those contradict your field of Public Health.  I understand your point and ok with that. You further mentioned, “more exhausting evaluation can be performed with further analyses.” Hence I recommend a minor revision.

  1. Following my comment:

“We need to think about whether the designs of the questionnaires are formulated to reach expected, predefined results or not. Though in the revision you included some limitations of this study, many relevant critical questionnaires are missing. At least those areas need to be discussed at some point in the text; otherwise, the conclusion adds little value addition to our existing knowledge.”

Your answer is “We as the authors are sad to hear you feel the article is scientifically unvaluable…“ Though my comment did not indicate it is scientifically invaluable. Some critical analyses and discussions always improve the qualities of a paper.  

Do you think those main results you obtained for COVID-19 vaccination hesitancy for South Carolina will be very different if you do the same analyses for other states of the US? Will the results be different if the analyses are conducted in another country, say UK?  Also, would your outcome be different if you consider participants who are vaccine-hesitant rather than COVID vaccine-hesitant?

Whether the perception of lower vaccination rate in South Carolina compared to other US states could also be related to age distribution (less old people), high/low qualified, income status, population density or number of highly populated major cities etc. – those may be mentioned. For comparison with other states of the US, those can also influence results.

  1. In my last two reviews, you did not attend one of my comments.

In your answer to my first review comment (2nd last reply), you mentioned: “The bold values are for those loadings that are greater than 0.40 as stated in the methods. If these variables did not result in an influence for the PCA analysis they were not bolded. The results of the analysis were” – you did not complete the sentence. Hence there is no reply to that comment.

Now in response to the 2nd review for the same comment, you mentioned “We apologize for the incomplete response. This has been corrected.”  - where is the response? I can not see your response anywhere.

  1. You mentioned in your response “the rise of recent variants occurred in vulnerable unvaccinated persons and populations. https://time.com/6124436/omicron-vaccine-inequity/”

The link is not from any published paper and moreover, it is published on November 29th, 2021. In a first moving situation, we now have many more observed facts for the last three months (Dec-Feb). Numerous people even after the third dose got infected with Omicron. Many third-vaccinated people who took all vaccine doses (1 and 2) on time, were infected with Omicron though also previously infected with other variants.  There are no proof from government data record that recent variant occurred among unvaccinated persons and the difference between vaccinated and unvaccinated are statistically different. It also contradicts the rise in cases for Gibraltar (100 % vaccination for a long time) and for a huge rise in transmission for many highly vaccinated countries.

People within two weeks of vaccination are more prone to have the disease, compared to the next few weeks (Keehner et al., 2021), though those groups of people are often categorized as unvaccinated or below category. Those could lead to many discrepancies in calculations.    

Ref:

Keehner et al. 2021, SARS-CoV-2 Infection after Vaccination in Health Care Workers in California, N. Engl. J. Med., doi: 10.1056/NEJMc2101927.March 23, 2021.

The authors did lots of work and they further acknowledged that some important variables in the overall COVID-19 vaccine hesitancy dilemma were not attended here. They also mentioned, “Vaccine hesitancy is complex and a more exhausting evaluation can be performed with further analyses.”

Hence, I am ok with that. I recommend publishing after minor revisions.

I do not want to review it again. 

Author Response

Thank you Reviewer 3 for your continued comments. Please find our succinct response below.

1-We have added the requested reference (#33): lines 491-492.

2-We added this additional limitation: lines 536-537.

3-We have corrected this sentence: lines 133-135.